# Dynamic brain connectivity predicts emotional arousal during naturalistic movie-watching

Jin Ke[1,2]*, Hayoung Song[1,3], Zihan Bai[1,4], Monica D. Rosenberg [1,5,6],
Yuan Chang Leong [1,5,6]*

1 Department of Psychology, Neuroscience Institute, The University of Chicago, Chicago, Illinois, United States of America, 2 Department of Psychology, Yale University, New Haven, Connecticut, United States of America, 3 Department of Neuroscience, Washington University School of Medicine, St Louis, Missouri, United States of America, 4 Department of Psychiatry, Yale School of Medicine, New Haven, Connecticut, United States of America, 5 Institute of Mind and Biology, Neuroscience Institute, The University of Chicago, Chicago, Illinois, United States of America, 6 Neuroscience Institute, The University of Chicago, Chicago, Illinois, United States of America

* jin.ke@yale.edu, ycleong@uchicago.edu

## Abstract

Human affective experience varies along the dimensions of valence (positivity or negativity) and arousal (high or low activation). It remains unclear how these dimensions are represented in the brain and whether the representations are shared across different individuals and diverse situational contexts. In this study, we first utilized two publicly available functional MRI datasets of participants watching movies to build predictive models of moment-to-moment emotional arousal and valence from dynamic functional brain connectivity. We tested the models by predicting emotional arousal and valence both within and across datasets. Our results revealed a generalizable arousal representation characterized by the interactions between multiple large-scale functional networks. The arousal representation generalized to two additional movie-watching datasets with different participants viewing different movies. In contrast, we did not find evidence of a generalizable valence representation. Taken together, our findings reveal a generalizable representation of emotional arousal embedded in patterns of dynamic functional connectivity, suggesting a common underlying neural signature of emotional arousal across individuals and situational contexts. We have made our model and analysis scripts publicly available to facilitate its use by other researchers in decoding moment-to-moment emotional arousal in novel datasets, providing a new tool to probe affective experience using fMRI.

## Author summary

This study explores how the brain represents two key dimensions of emotional experience: valence (how positive or negative an experience feels) and arousal (the level of emotional activation). Using publicly available brain imaging data

**Data availability statement:** All data are available in the main text or the supplementary materials. The behavioral and brain data, trained model weights, the analysis code, as well as a step-by-step instruction to run the analysis code, are openly available on GitHub: https://github.com/jinke828/AffectPrediction

**Funding:** This work was supported by the National Science Foundation (BCS-2043740 to M.D.R.). The funders had no role in study design, data collection and analysis, decision to publish, or preparation of the manuscript.

**Competing interests:** The authors have declared that no competing interests exist.

from people watching movies spanning different genres, storylines, and characters, we built computational models to predict moment-to-moment changes in emotional valence and arousal from patterns of brain connectivity. Testing these models across datasets, we identified a common set of brain connections that capture a shared neural signature of emotional arousal across different individuals and movies. These results suggest that the brain may represent emotional arousal in a consistent manner across people and situations. However, we did not find evidence for a similarly generalizable neural pattern for emotional valence. By sharing our model and analysis tools openly, we aim to support other researchers in using brain imaging to better understand emotional experiences in naturalistic, real-world settings. Our findings contribute to a growing body of research focused on uncovering how emotional experiences are represented in the brain.

## Introduction

Human experience is characterized by a continual ebb and flow of emotions, shaping our attention [1,2] and memory [3,4], as well as guiding how we make decisions and interact with others [5–8]. These affective experiences, though diverse and complex, are thought to be organized along a small number of principal dimensions [9–11]. Among these, valence and arousal are two dimensions that are central to many contemporary theories of emotion [12–15]. Valence refers to the positivity or negativity of an emotional state, while arousal refers to its intensity or activation level. Emotional states can be placed in a two-dimensional space based on their level of valence and arousal—for example, both "excitement" and "anger" are high in arousal but occupy opposite ends on the valence dimension [16]. Thus, this framework provides a useful model for illustrating how seemingly distinct emotional experiences share underlying similarities and relate to one another in a structured space. A recent study found that individuals' positions within this space in a given situation predicts their behavior during social interactions [17], suggesting that these dimensions not only describe emotional states but also serve as reliable indicators for predicting behavioral responses.

The concept of valence and arousal as core components of affect has significantly influenced research and theories in emotion [18–20]. In particular, this perspective proposes an underlying similarity between excitement and anger due to both being high arousal experiences. Similarly, joy and gratitude are thought to be similar due to both being positively valenced. However, the extent to which these similarities extend to how the brain represents affective experiences remains an open question. Specifically, are there generalizable neural patterns associated with valence or arousal across various contexts? Moreover, given proposals that valence and arousal are processed by core neural systems [20,21], are these representations shared across individuals?

Prior research has identified several brain regions where activity correlates with valence and arousal, including the orbitofrontal cortex (OFC), amygdala, ventral

striatum, and insula [22–30]. More recently, there is growing evidence that patterns of dynamic connectivity between brain regions may also contain information about affective experience [31–34]. For example, a recent study by Young and colleagues [34] presented participants with a stressful movie scene and showed that connectivity between the salience network was associated with fluctuations in heart rate, a measure of physiological arousal. However, this study focused on a single movie stimulus, leaving it unclear whether the same connectivity-arousal relationship would generalize to other stimuli with different situational contexts. Moreover, it remains uncertain whether the same networks would be engaged when probing "emotional arousal", where individuals have subjective awareness of their level of arousal and can report them behaviorally [35].

The present study aims to determine whether generalizable neural patterns underlie emotional valence and arousal across complex, naturalistic stimuli. This work extends prior research in three significant ways. First, instead of focusing on specific brain regions or networks, we trained our models on whole-brain functional connectivity dynamics. Specifically, we used the dynamic connectome-based predictive modeling (CPM) approach developed by Song et al. [36,37] to predict moment-to-moment ratings of emotional valence and arousal during movie-watching, and assessed whether the models generalized across different movies. CPMs are so named because they leverage the connectome (i.e., the comprehensive map of functional connections in the brain) to make predictions about inter- or intra-individual differences in behavior or mental states [38].

This approach builds on the growing recognition that whole-brain interactions play an important role in facilitating diverse cognitive processes [39–41], with studies showing that whole-brain functional connectivity patterns predict both intra-individual fluctuations and inter-individual differences in attention [36,42], memory [43,44], and narrative comprehension [37]. Affective experience involves the integration of sensory inputs, past experiences and current goals, which requires the coordination between networks with different functions [45]. Given that affective experience engages multiple brain regions simultaneously, a whole-brain connectivity approach may be particularly well-suited to capture the dynamic and distributed nature of the underlying brain states.

Second, previous studies have often relied on brief, decontextualized stimuli, such as isolated visual images or words, to investigate these affective dimensions [22–28,46]. Such stimuli may not capture the diversity and complexity of real-world emotional experiences, raising questions about whether the neural representations of valence and arousal identified under these controlled conditions would generalize across varied situational contexts and individuals. Movies, by contrast, provide extended, context-rich narratives that elicit dynamic and temporally continuous emotional responses [36,47,48]. While watching a movie is a passive, second-person experience and does not encompass the full range of real-life emotional contexts, they serve as a valuable step toward understanding affective experiences in more ecologically valid settings, where situations are complex, dynamic, and varied. Here, we examined the neural activity associated with continuous self-reported affective experiences during full-length TV episodes, allowing us to capture moment-to-moment affective responses across a variety of situations.

Third, we assessed the out-of-sample generalizability [49] of our findings by testing our models on new stimuli, participants, and datasets that were not used to train the model. This step is crucial to ascertain whether our models were fitting to the idiosyncrasies of specific contexts and a group of individuals, or if they genuinely reflect generalizable neural representations of valence and arousal. For example, a model that predicts arousal in a given movie may have learned to associate arousal with the appearance of a specific main character during arousing scenes. Such a model would fail when applied to a different movie with different characters and plotlines. In contrast, a model that captures neural patterns shared across various arousing situations would be expected to generalize successfully to new movie stimuli.

We utilized two publicly available fMRI datasets of participants watching TV episodes ($N = 16$ [47] and $N = 35$ [48]; Fig 1), and collected continuous valence and arousal ratings of the two episodes from a separate group of participants ($N = 120$, 30 for each pairing of affective rating and movie–e.g., arousal ratings for *Sherlock*). We first verified that the ratings were consistent across participants watching the same movie, indicating that the average group ratings provide

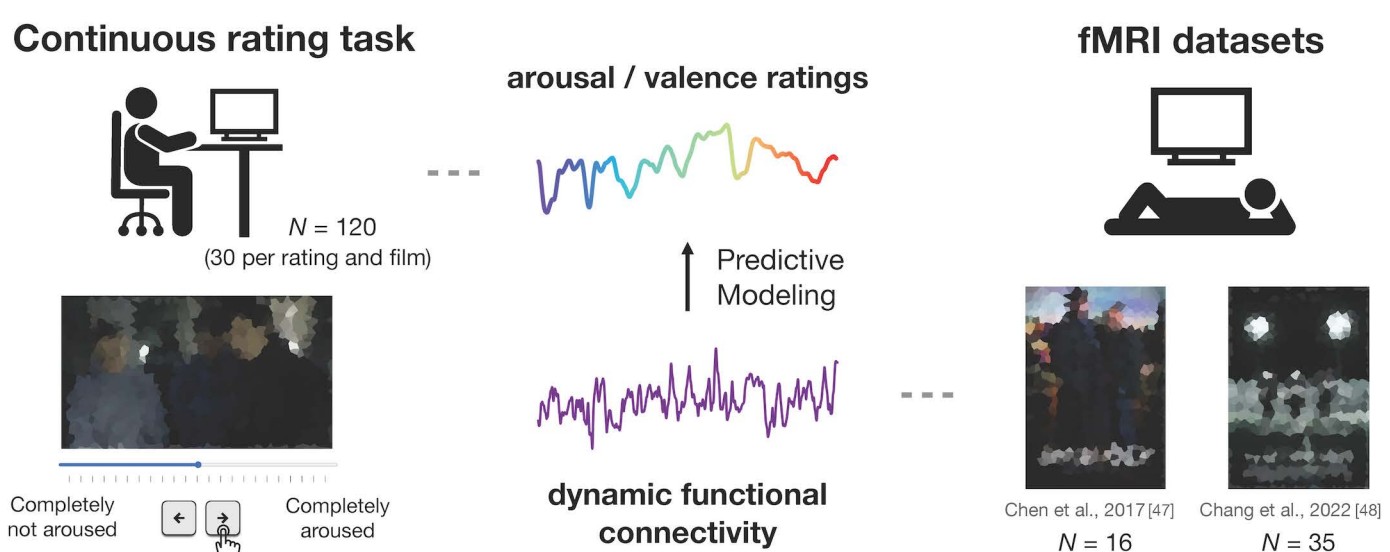

**Fig 1. Study schematic procedure.** We utilized two publicly available fMRI datasets of participants watching *Sherlock* (N = 16) [47] and *Friday Night Lights* (N = 35) [48]. Additionally, we collected behavioral ratings of the two episodes from a separate group of participants. Each participant is presented with one movie clip and is instructed to continuously rate how the clip is making them feel in terms of either valence (i.e., positive to negative) or arousal (i.e., not aroused to aroused). The 120 participants in the rating study were equally divided between conditions. Thus, there were 30 participants per pairing of an affective dimension (valence or arousal) and movie (*Sherlock* or *Friday Night Lights*). We blurred the Images in the figures for copyright reasons; in the experiment, movies were shown at high resolution. We built within and across dataset CPMs to predict moment-to-moment affect ratings from dynamic whole-brain functional connectivity patterns.

a reliable proxy for individual affective experience. Next, we built CPMs to predict valence and arousal fluctuations from dynamic functional connectivity patterns. We assessed both the within-dataset accuracy (i.e., how well the CPM predicted valence or arousal of the held-out participant in the dataset it was trained on) as well as across-dataset accuracy (i.e., how well a CPM predicted valence or arousal in the dataset it was not trained on).

To further validate the robustness and generalizability of our results, we tested the trained CPMs on two additional fMRI datasets of different participants watching different movies ($N = 18$ [50] and $N = 32$, new dataset collected by our group). Altogether, our approach seeks to identify generalizable neural representations of moment-to-moment valence and arousal based on the interaction between brain areas, and shed new light on the neural activity underlying affective experience.

## Results

### Affective experience is synchronized across individuals during movie watching

One-hundred and twenty participants performed an affect rating task while watching one of two TV episodes. One was a 48-min long episode from BBC's *Sherlock* (a British mystery crime drama series) while the other was a 45-min long episode from NBC's *Friday Night Lights* (an American sports drama series). Each participant provided continuous ratings of either valence (i.e., positive to negative) or arousal (i.e., not aroused to aroused) while watching one of the videos ([Fig 1]). As we were primarily interested in subjective feelings during the movie, we instructed participants to indicate their affective experience (e.g., how positive or negative they were feeling), rather than the perceived emotionality of the movies (e.g., whether they thought the current scene was positive or negative). Previous work found that providing continuous affect ratings did not alter emotional or neural responses to emotion-eliciting films, indicating that this approach can capture participants' affective experiences without disrupting their natural responses to the content [51]. We will refer to the pairing of an affective dimension and a movie as an "experimental condition" (e.g., *Sherlock*-valence). Thus, we have four

experimental conditions with a sample size of 30 participants each. The group-averaged time courses were treated as a proxy of the affective experience time-locked to the movie (Fig 2A).

We first examined whether subjective ratings of valence and arousal were synchronized across individuals during movie watching. We computed the Pearson correlation between each individual's rating and the group-averaged rating with the individual left out. The average intersubject correlation was significantly positive across all conditions (Fig 2B; *Sherlock*-valence: mean $r = .700$, *s.d.* $= .197$, $p < .001$; *Sherlock*-arousal: mean $r = .502$, *s.d.* $= .196$, $p < .001$; *Friday Night Lights*-valence: mean $r = .653$, *s.d.* $= .167$, $p < .001$; *Friday Night Lights*-arousal: mean $r = .787$, *s.d.* $= .109$, $p < .001$). Statistical significance was assessed with a nonparametric permutation test that accounts for the autocorrelation structure

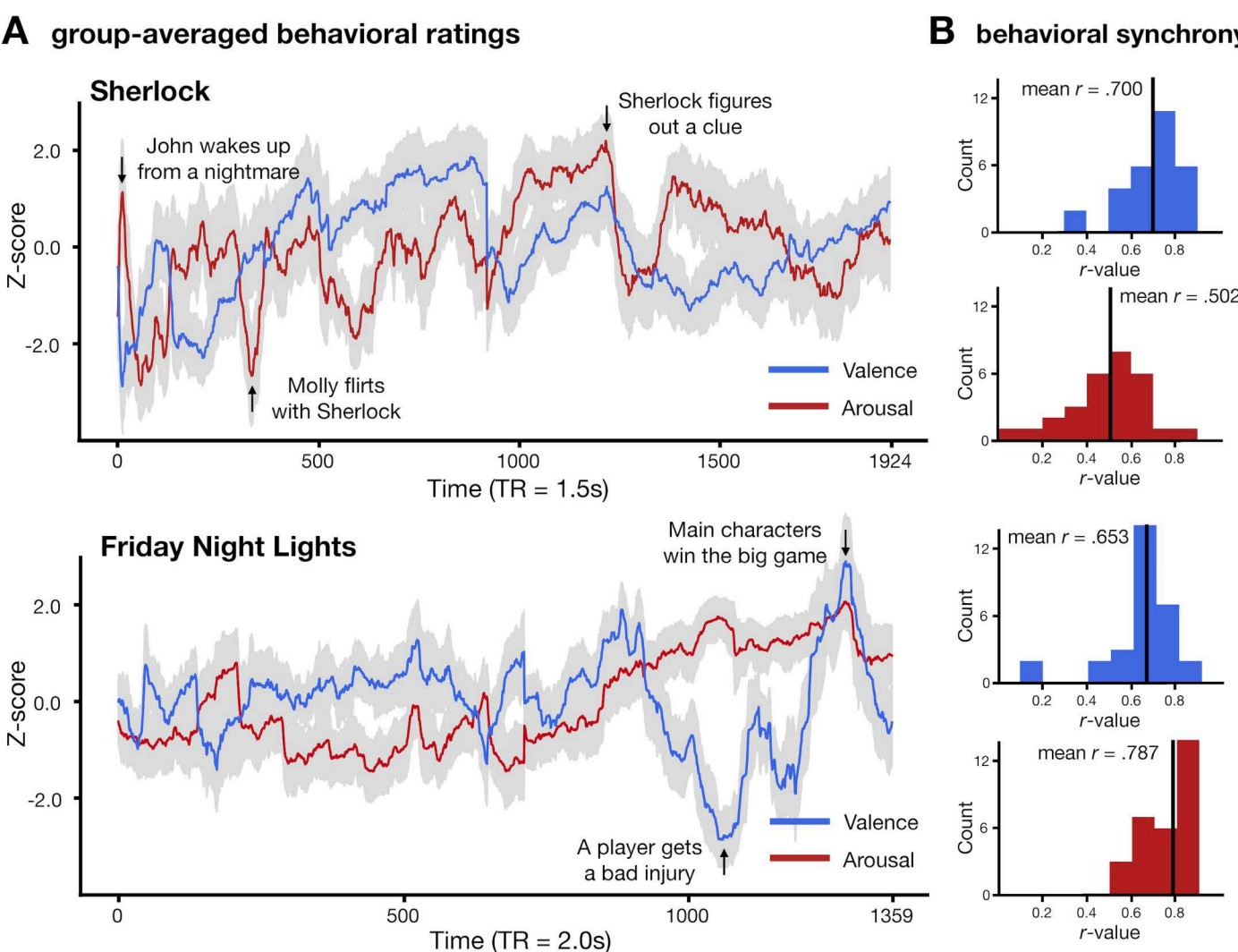

**Fig 2. Affective experience is synchronized across participants in all four experimental conditions.** A. Participants' subjective affective experience fluctuates over time during naturalistic movie watching. The red and blue solid lines indicate average arousal and valence time courses respectively. The gray areas indicate the standard deviation of ratings across participants at each time point. Each condition includes 30 participants. B. Histograms of the similarity of each individual's subjective ratings to the group-averaged rating with the individual left out. Higher mean *r* values indicate stronger affective synchrony.

of the timeseries [52]. These results suggest that affective experience, across both valence and arousal dimensions, was synchronized across individuals when they watched both TV episodes.

The average valence and arousal time courses were not significantly correlated in both *Sherlock* ($r = .094$, $p = .359$) and *Friday Night Lights* ($r = -.224$, $p = .187$). The average time courses of arousal and the absolute value of valence were significantly correlated in *Friday Night Lights* ($r = .706$, $p < .001$) but not *Sherlock* ($r = .101$, $p = .303$). Qualitatively, the average valence and arousal time courses reflected expected fluctuations in affective experience during the episodes (Fig 2A). For example, while watching *Sherlock*, participants' ratings indicated high negative valence and high arousal during the scene where John Watson has a nightmare of his time in the war, and high positive valence but low arousal during a scene where Sherlock was clueless to the lab assistant, Molly, flirting with him. Similarly, while watching *Friday Night Lights*, participants' ratings indicated high negative valence and high arousal when a star player is severely injured in a football game, and high positive valence and high arousal when the main character's team wins the game. Having demonstrated the consistency of valence and arousal time courses across participants watching the same TV episode, we next sought to examine whether fluctuations in valence and arousal could be predicted from whole-brain dynamic connectivity.

### Dynamic functional connectivity predicts emotional arousal within and across datasets

We first asked whether fluctuations in emotional arousal could be predicted from dynamic functional connectivity. To that end, we trained CPMs to predict arousal rating time courses from time-resolved dynamic functional connectivity (FC) patterns (Fig 1). We parcellated the whole brain into 122 ROIs, following the Yeo atlas for cortical regions [53] (114 ROIs) and the Brainnetome atlas for subcortical regions [54] (8 ROIs: bilateral amygdala, hippocampus, basal ganglia, and thalamus). The dynamic FC patterns were extracted from the 122-ROI-based BOLD time courses using a tapered sliding window approach, where the Fisher's z-transformed Pearson's correlation between the BOLD time courses of every pair of ROIs was computed within each tapered sliding window [36] (window size - *Sherlock*: 30 TRs = 45s; *Friday Night Lights*: 23 TRs = 46s; see Methods).

Separate models were trained on two openly available fMRI datasets where participants watched *Sherlock* [47] (*N* = 16) or *Friday Night Lights* [48] (*N* = 35). We assessed model performance in predicting group-mean arousal ratings within each dataset as well as between datasets. *Within-dataset performance* was computed using a leave-one-subject-out cross-validation approach, training the model on the neural data of all but one participant in a dataset, and applying the trained model to data from the held-out participant to predict the average arousal time course. In every round of cross-validation, we performed a feature selection step where we selected FCs that significantly correlated with the arousal time course in the training set of N - 1 subjects (one-sample Student's t test, $p < .01$), following procedures described in prior work [36,38,42,55]. The model was then tested on the held-out subject. Neither the model training nor the feature selection step involved the held-out subject, ensuring no data leakage. Model accuracy was computed as the average correlation between the model-predicted arousal time course and the empirical arousal time course across cross-validation folds. As a second measure of model performance, we computed the root mean squared error (RMSE) between model predictions and the behavioral ratings.

Statistical significance was assessed by comparing model accuracy against a null distribution of 1000 permutations generated by training and testing the model on phase-randomized behavioral ratings. For a given permutation iteration, the model was trained and tested on the same phase-randomized time course. In other words, the model can potentially learn arbitrary features correlated with that iteration's phase-randomized time course and use those for prediction during testing. The resulting null distribution thus provides a conservative test of model performance when training and testing within the same dataset, as it reflects the baseline performance expected if the model learned arbitrary features associated with a stimulus. Indeed, the observed null distribution was positively skewed (centered around 0.5 rather than 0; Fig 3A, left), suggesting that the null models learned associations between FC connections and arbitrary stimulus features that correlated with the phase-randomized behavioral time course in each iteration.

Relative to this conservative null, within-dataset accuracy was significantly above chance in predicting arousal ratings for both *Sherlock* (mean *r*=.575, *p*=.034; RMSE=.825, *p*=.034) and *Friday Night Lights* (mean *r*=.734, *p*=.002; RMSE=.701, *p*<.001; Fig 3A, left), indicating that moment-to-moment experience of emotional arousal could be decoded from whole-brain FC patterns in both movies. However, an alternative possibility was that the CPM was learning features specific to the particular movie it was trained on rather than arousal per se. For example, if Sherlock Holmes tended to appear in scenes that were highly arousing, the model could learn to associate the neural representation of Sherlock Holmes with higher arousal ratings. In a different movie, where Sherlock Holmes was not present, or no longer associated with arousing scenes, the model would fail to predict arousal. Thus, it would be important to test if the model would generalize to a novel context.

With that goal in mind, we assessed the *across-dataset performance* to test whether the models generalized across the two movies. For each dataset, we first identified the set of FC features that were selected in every round of the cross-validation, which we term the arousal network for that dataset. To test model generalizability, we trained a CPM in one dataset on its arousal network to predict the group-average arousal time course and applied the trained model to the other dataset with a different group of participants watching a different movie. A null distribution was again generated by training and testing on phase-randomized behavioral ratings. However, in this case, the training and test data consists of different stimuli, thus any non-specific associations learned by the model from the training dataset would not generalize to the test dataset. As expected, the null distribution was centered around zero (Fig 3A, right).

The across-dataset predictions of arousal (Fig 3A, right) were significantly above chance both for a model trained on *Sherlock* and tested on *Friday Night Lights* (mean *r*=.270, *p*=.010; RMSE=.977, *p*=.009), and a model trained on *Friday Night Lights* and tested on *Sherlock* (mean *r*=.198, *p*=.001; RMSE=.999, *p*=.001). These results indicate that our models generalized across movies and were not merely overfitting to a single movie.

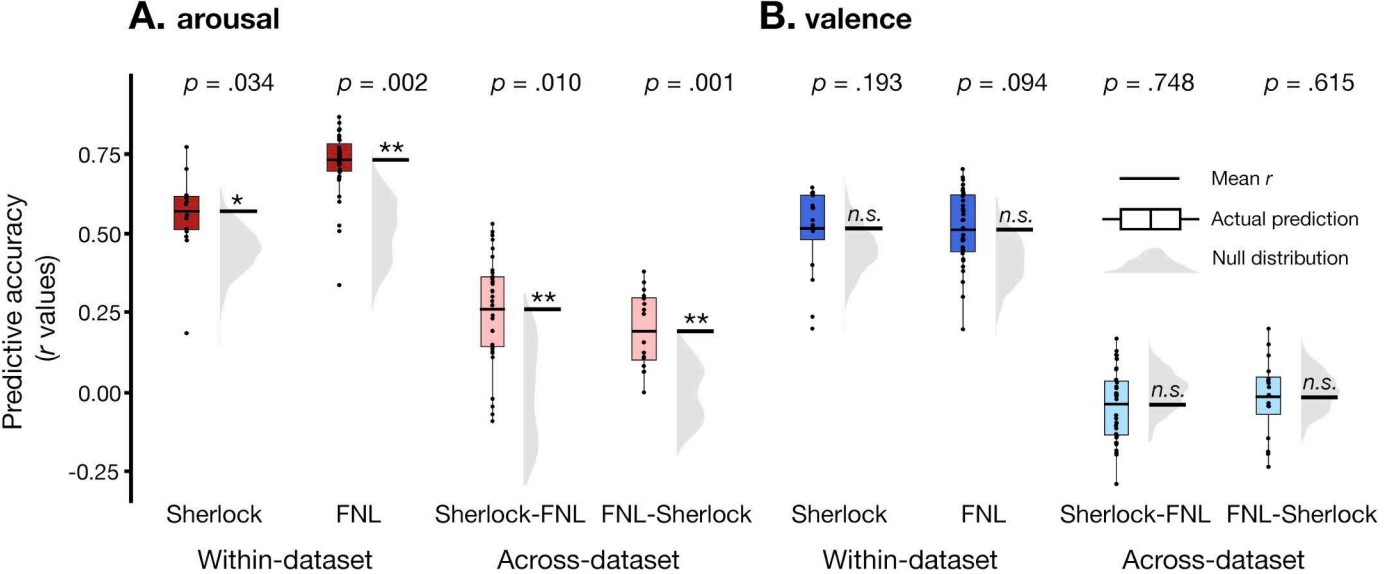

**Fig 3. Dynamic functional connectivity predicts fluctuations in arousal but not valence.** CPM performance in predicting arousal (**A**) and valence (**B**) within-dataset (the left panel) and between-dataset (the right panel). The y-axis represents the predictive accuracy, as measured by Pearson's correlation between the model predicted time course and the observed group-average time course. Each datapoint in the box plot represents the predictive accuracy of each round of cross-validation. The black horizontal lines show the mean *r* value computed from the average of the Fisher-*z* transformed individual-subject *r* values. The gray half-violin plots show the null distribution of 1000 permutations, generated by phase-randomizing the observed group-average time course before training and testing the models. *\*p*<0.05, *\*\*p*<0.01, n.s.: p>0.05, as assessed by comparing the empirical mean predictive accuracy against the null distribution.

To assess whether prediction results were driven by low-level audio-visual features, we reran the prediction analyses after regressing out ten low-level features (hue, saturation, pixel intensity, motion energy, presence of a face, whether the scene was indoor or outdoor, presence of written text, amplitude, pitch, presence of music) from each participant's BOLD signal time courses. These features were selected following the procedures prior studies examining the neural correlates of high-level cognitive processes using dynamic audio-visual stimuli [56,57]. Consistent with our previous analyses, we found above-chance predictions when the models were tested within each dataset (*Sherlock*: mean $r = .558$, $p = .024$; RMSE $= .834$, $p = .025$; *Friday Night Lights*: mean $r = .732$, $p = .003$; RMSE $= .701$, $p = .002$) and across datasets (train on *Sherlock* test on *Friday Night Lights*: mean $r = .267$, $p = .012$; RMSE $= .979$, $p = .011$; train on *Friday Night Lights* test on *Sherlock*: mean $r = .220$, $p = .020$; RMSE $= .990$, $p < .001$), suggesting that prediction accuracy was not confounded by these low-level features.

Additionally, to examine whether motion influenced the CPM predictions on arousal, we re-ran the analyses after regressing out each individual's frame-to-frame displacement from their dynamic functional connectivity time courses. Consistent with our previous results, the model significantly predicted arousal in both *Sherlock* (mean $r = .571$, $p = .028$.; RMSE $= .827$, $p = .030$) and *Friday Night Lights* (mean $r = .641$, $p = .004$; RMSE $= .790$, $p = .015$). The across-dataset predictions were similarly significantly above chance both when trained on *Sherlock* tested on *Friday Night Lights* (mean $r = .238$, $p = .015$; RMSE $= .990$, $p = .025$) and trained on *Friday Night Lights* tested on *Sherlock* (mean $r = .151$, $p = .034$; RMSE $= 1.007$, $p = .036$). These results suggest that our results were unlikely to be driven by motion artifacts.

Thus far, our results relied on a measure of dynamic FC computed within a 45s sliding window. To examine if and how prediction accuracy varied with the size of the sliding window, we assessed the across-dataset model performance with dynamic FC computed with window sizes of 15s, 30s, 60s, and 75s. Across all window sizes, across-dataset prediction accuracy of the model was significantly above chance (S2 Fig, all $p < .05$), indicating that our results were robust to different window sizes. Altogether, these analyses indicate that moment-to-moment fluctuations in arousal can be decoded from dynamic FC patterns, and that our results are robust to analytical decisions and low-level confounds.

## FC patterns within and between functional networks predict emotional arousal

Do the predictive models of arousal rely on information present in FCs within specific functional networks (e.g., dorsal attention network, default mode network), or is that information distributed across interactions between multiple networks? To answer this question, we examined the arousal network in each dataset. We term the FCs that were positively or negatively correlated with the arousal time course as positive or negative features respectively. The *Sherlock* arousal network includes 593 FC features (439 positive and 154 negative), and the *Friday Night Lights* arousal network includes 1578 FC features (848 positive and 730 negative).

We defined the set of overlap FC features between the *Sherlock* positive arousal network and the *Friday Night Lights* positive arousal network as the high-arousal network, and the set of overlap FC features between the *Sherlock* negative arousal network and the *Friday Night Lights* negative arousal network as the low-arousal network. We then assessed the degree of overlap in the high- or low-arousal networks across datasets. Both the high-arousal network (169 overlapping FC features, $p < .001$) and low-arousal network (68 overlapping FC features, $p < .001$) have an above-chance number of overlapping FC features across the two datasets. In contrast, the *Sherlock* high-arousal network does not significantly overlap with the *Friday Night Lights* low-arousal network (1 overlapping FC feature, $p > .99$), and the *Friday Night Lights* high-arousal network does not significantly overlap with the *Sherlock* low-arousal network (4 overlapping FC features, $p > .99$). Statistical significance of network overlap was assessed both using a hypergeometric cumulative density function [58] and using the Jaccard index as the measure of network similarity [59] (see Methods). Both methods generated consistent results.

To further understand how connections within and between large-scale functional networks constitute the high- and low-arousal network, we followed Yeo et al. [53] to group the 122 ROIs into 8 canonical functional networks, namely, the

PLOS Computational Biology

visual (VIS), somatosensory-motor (SOM), dorsal attention (DAN), ventral attention (VAN), limbic (LIM), frontoparietal control (FPN), default mode (DMN), and subcortical (SUB) networks. We asked whether particular functional networks were represented in the arousal network more frequently than chance ([Fig 4A]). We computed the proportion of selected FCs among all possible FCs between each pair of functional networks, and assessed the significance of the proportions by

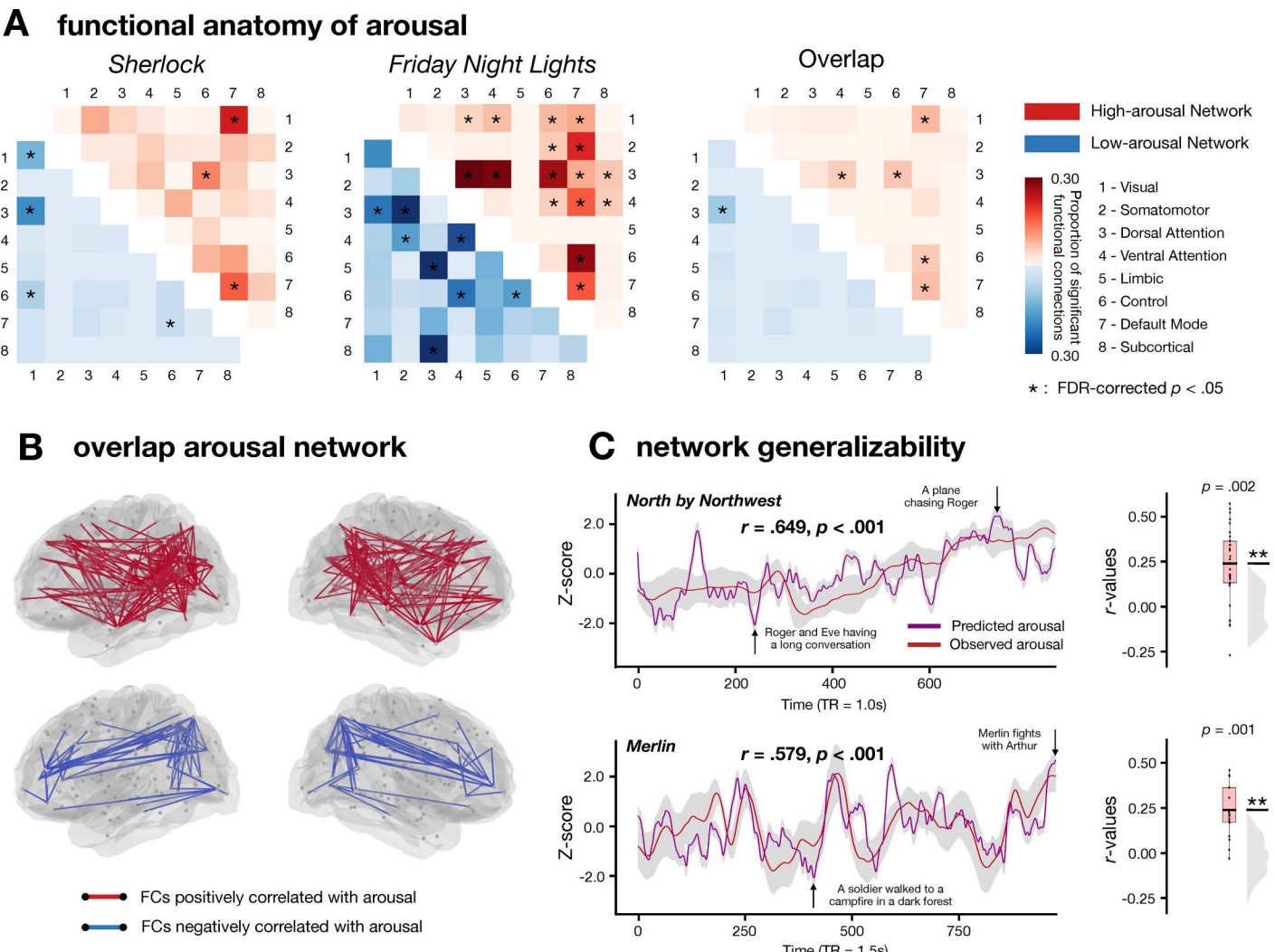

**Fig 4. Functional anatomy of arousal.** A. Arousal networks for *Sherlock*, *Friday Night Lights*, and their overlap. Each cell represents the proportion of selected FCs among all possible FCs between each pair of functional networks. The cells in the upper triangle represent the high-arousal network (red), and the cells in the lower triangle represent the low-arousal network (blue). Network pairs with above-chance selected FCs are indicated with an asterisk (one-tailed *t*-test, FDR-corrected $p < .05$). B. Visualization of the overlap arousal network. Each node represents a brain region. The lines connecting two nodes show their functional connection (red: high-arousal network; blue: low-arousal network). C. Connectome-based model trained on the overlap arousal network generalized to two more fMRI datasets, *Merlin* and *North by Northwest*. The left figure indicates how well the model could predict the group averaged experiences of arousal. We averaged the model predictions across participants watching the same movie and computed the correlation between the group-average model predictions (predicted arousal) and group-average arousal ratings from a separate group of individuals (observed arousal). The gray bands indicate the standard deviation of predicted arousal across participants at each time point. The right figure shows the predictive accuracy of the across-dataset prediction. The model was trained on a minimal set of FCs that predicted arousal in both *Sherlock* and *Friday Night Lights*, and tested separately on *Merlin* and *North by Northwest*. **$p < .01$, as assessed by comparing the empirical mean predictive accuracy against the null distribution.

comparing them against a null distribution of 10000 permutations where functional connections were randomly distributed across networks (see Methods). Connections between the DMN and FPN, DAN and FPN, and DAN and VAN, as well as connections within the DMN were positively associated with arousal. In contrast, connections between the DAN and VIS network were negatively associated with arousal (FDR-corrected $p < .05$). These results suggest that information about emotional arousal is decoded from functional connections both within the DMN and between pairs of distributed networks.

### Examining the relationship between emotional arousal and attentional engagement

A recent study analyzing the *Sherlock* dataset revealed that self-reported attentional engagement is correlated with scene-by-scene annotations of emotional arousal during narratives. This previous study characterized a set of functional networks that predicts engagement across datasets [36]. The average engagement rating from that study was significantly correlated with the average arousal ratings that we collected (Pearson's $r = .778$, $p = .014$) but not the average valence ratings (Pearson's $r = .085$, $p = .404$). Relatedly, the arousal and engagement networks within the *Sherlock* dataset significantly overlap (361 overlapping FC features, $p < .001$; no overlap between the high arousal/ low engagement and low arousal/high engagement networks). The shared FC features constitute 60.88% of FCs in the arousal network and 52.79% of FCs in the engagement network, and were distributed across different functional networks (S4 Fig).

However, model-predicted engagement was not significantly correlated with behavioral arousal ratings (mean $r = .498$, $p = .173$; MSE $= .754$, $p = .163$). Moreover, this correlation was significantly lower than that between model-predicted arousal and behavioral arousal ratings ($r = .498$ vs. $.558$; paired-*t* test, $t(15) = -4.368$, $p < .001$). Likewise, model-predicted arousal ratings were not significantly correlated with behavioral engagement ratings (mean $r = .518$, $p = .172$; MSE $= .735$, $p = .167$), and this correlation was significantly lower than that between model-predicted engagement ratings and behavioral engagement ratings ($r = .518$ vs. $.572$; paired-*t* test, $t(15) = -4.115$, $p < .001$). In other words, even though behavioral ratings of engagement and arousal were correlated, we found a double association such that model-predicted arousal was associated with behavioral arousal ratings, and model-predicted engagement was associated behavioral engagement ratings, but neither model accurately predicts the other measure. These results suggest that attentional engagement and arousal are related but non-synonymous constructs, with arousal possibly comprising an aspect of attentional engagement [60,61].

### Connectome-based models of arousal generalized to two more fMRI datasets

Our findings that CPMs trained on one movie predicted arousal in another largely distinct movie provide evidence of a generalizable neural representation of arousal that can be decoded from dynamic functional connectivity. We then proceeded to test whether our identified arousal network would generalize to additional datasets. To that end, we analyzed two additional fMRI movie datasets with distinct genre, storyline, characters, and duration. The first was a 15-min clip from the movie *North by Northwest* (a spy thriller directed by Alfred Hitchcock; $N = 32$) collected by our group, and the second was a 25-min episode from BBC's *Merlin* (a British fantasy adventure drama series; $N = 18$) from Zadbood and colleagues [50]. To measure affective experience during the movie clips, we collected behavioral ratings of arousal and valence for both movies ($N = 60$, 15 in each condition).

We then trained a CPM on both *Sherlock* and *Friday Night Lights* datasets with the overlap arousal network (169 positive FCs and 68 negative FCs; Fig 4B) as input features. We then tested the model separately on the *North by Northwest* and *Merlin* datasets. Prediction accuracy was above-chance in both *North by Northwest* (mean $r = .227$, $p = .002$; RMSE $= .985$, $p < .001$) and *Merlin* (mean $r = .230$, $p = .004$; RMSE $= .990$, $p < .001$; Fig 4C, right). Model-predicted arousal time courses also corresponded with the plot of each movie. For example, in *North by Northwest*, the most arousing moment predicted by the model occurred when the protagonist was being chased by a plane, while the least arousing moment occurred during a long conversation between characters (S5 Fig, top). Similarly, in *Merlin*, the most arousing

moment predicted by the model occurred when the two main protagonists had a brawl in a tavern, while the least arousing moment occurred when a nameless soldier walked towards a campfire (S5 Fig, bottom). These results indicate robust out-of-sample generalizability, suggesting that the models were not merely capturing the idiosyncrasies of specific datasets. Instead, the arousal networks that we identified generalized across diverse situations.

In the prior analyses, we correlated the model predictions from each participant's data with the group-average arousal ratings, and then averaged the correlations across the sample. This allowed us to assess the extent to which predictions from each individual matched the group average. If our goal was instead to maximize our ability to predict the group-average experience of arousal, we can first aggregate the model predictions across participants and predict average arousal ratings from the aggregated model predictions. To that end, we averaged the model predictions across participants watching the same movie and computed the correlation between the group-average model predictions and group-average arousal ratings. The group-average predicted arousal time course was significantly correlated with the group-average human-rated arousal time course in both *North by Northwest* ($r = .649$, $p < .001$) and *Merlin* ($r = .579$, $p < .001$; Fig 4C, left). Similarly, we observed a significant correlation between the group-average human-rated arousal and predicted time courses in both *Sherlock* ($r = .408$, $p < .001$) and *Friday Night Lights* ($r = .621$, $p < .001$; S6 Fig) datasets. These results provide a measure of the model's ability to predict the average arousal time course from a different group of participants watching a movie that the model was not trained on, and suggest that our model could be used to generate moment-to-moment predictions of arousal during movie watching in new, independent datasets.

## Dynamic functional connectivity does not predict moment-to-moment valence

Having established that emotional arousal can be predicted from dynamic functional connectivity, we examined whether the same approach can predict valence. Within-dataset CPMs did not show above-chance predictive accuracy in either *Sherlock* (mean $r = .518$, $p = .193$; RMSE = .860, $p = .195$) or *Friday Night Lights* (mean $r = .514$, $p = .094$; RMSE = .861, $p = .090$). Across-dataset predictive accuracy was also not significantly better than chance (train on *Sherlock* test on *Friday Night Lights*: mean $r = -.041$, $p = .748$; RMSE = 1.097, $p = .789$; train on *Friday Night Lights* test on *Sherlock*: mean $r = -.016$, $p = .615$; RMSE = 1.085, $p = .812$; Fig 3B). Equivalence tests [62–64] indicated that these accuracies were statistically indistinguishable from zero within bounds defined by a minimal effect size of interest [-.100,.100] (see S7 Fig). One possible explanation of these null results is that the group-average valence ratings might not be sufficiently reliable across participants, potentially due to idiosyncratic experience of subjective valence. However, the inter-subject agreement of the valence ratings was not systematically different than that of arousal in both datasets and showed comparable overall inter-subject agreement across-datasets (valence: mean $r = .677$; arousal: mean $r = .645$; $t$-test: $t(118) = .756$, $p = .451$). These results suggest that the model's poor performance at predicting valence fluctuations were unlikely to be due to lower reliability of the behavioral valence ratings.

Another possible explanation is that the current specific window size we used in the predictive modeling did not capture the valence fluctuations on the right timescale. To rule out this possibility, we examined whether the valence model could generalize across datasets when the model was trained and tested on various different timescales. We used the same approaches as predicting arousal. None of the 25 combinations of window sizes showed above-chance accuracy in predicting valence across-dataset, suggesting that the null results were unlikely to be due to the choice of window size (S3 Fig).

The previous analyses assumed that valence is decodable as a single "bipolar" dimension, with positive and negative valence at opposing ends [12,20]. An alternative possibility is that positive and negative valence are represented separately in the brain [65–68], in which case, a CPM trained on data that spanned the entire valence spectrum would not perform well. To test this alternative possibility, we trained separate CPMs on positive and negative moments in each movie. For positive valence, we observed above-chance within-dataset predictive accuracy in *Friday Night Lights* (mean $r = .762$, $p = .001$; RMSE = .677, $p < .001$) but not Sherlock (mean $r = .431$, $p = .835$; RMSE = .907, $p = .820$). However, the model

failed to generalize across-dataset (from *Sherlock* to *Friday Night Lights*: mean $r$ = -.020, $p$ = .528; RMSE = 1.118, $p$ = .565; from *Friday Night Lights* to *Sherlock*: mean $r$ = .088, $p$ = .247; RMSE = 1.029, $p$ = .156; S8B Fig, left). For negative valence, neither within-dataset (*Sherlock*: mean $r$ = .553, $p$ = .400; RMSE = .839, $p$ = .386; *Friday Night Lights*: mean $r$ = .647, $p$ = .192; RMSE = .779, $p$ = .204) nor across-dataset (from *Sherlock* to *Friday Night Lights*: mean $r$ = -.093, $p$ = .875; RMSE = 1.125, $p$ = .958; from *Friday Night Lights* to *Sherlock*: mean $r$ = -.216, $p$ = .898; RMSE = 1.159 $p$ = .885; S8B Fig, right) models showed significant predictive accuracy. As such, we were unable to predict valence from dynamic functional connectivity, even when considering positive valence and negative valence separately.

### Connectome-based models of valence failed to generalize even with more training data

One reason why the valence CPMs failed to generalize may be because they were trained on insufficiently diverse contexts. We thus sought to test if a valence CPM would generalize if trained on more data. To that end, we trained a CPM on the combined data from *Sherlock* and *Friday Night Lights*, with the 2067 FCs that significantly correlated with valence in the combined dataset as input features. This model failed to generalize to both *Merlin* (mean $r$ = -.010, $p$ = .611; RMSE = 1.071, $p$ = ,659) and *North by Northwest* (mean $r$ = -.140, $p$ = .930; RMSE = 1.095, $p$ = .873), indicating that despite training the model on multiple datasets, we were still unable to predict valence from whole-brain dynamic functional connectivity.

## Discussion

This study utilized predictive modeling of affective experience to examine how valence and arousal states are represented in the human brain. We first showed that subjective behavioral ratings of valence and arousal were synchronized between individuals watching the same movie. We then identified a generalizable neural representation of emotional arousal across individuals and situational contexts that can be decoded from patterns of whole-brain dynamic functional connectivity. Specifically, a model trained to predict arousal from dynamic functional connectivity during one movie generalized to a second movie, and the arousal representation further generalized to two additional datasets with different groups of participants. In contrast to arousal, we were unable to predict valence from functional connectivity dynamics.

A strength of the current work is that, in addition to examining the neural correlates of affective experience in a particular context, we explicitly tested the out-of-sample generalizability across multiple independent datasets with different movie stimuli. This is crucial because predictive models trained and tested within the same context may merely fit the idiosyncrasies of a specific stimulus. In contrast, a generalizable neural representation of a latent psychological construct should be consistent across various contexts and situations [69–73]. Our results indicate that the neural representation of arousal decoded from dynamic functional connectivity meets this criterion of generalizability—the CPMs successfully predicted fluctuations in arousal in movies that the models had not previously encountered, which also differed in low-level audio-visual features, characters, plot, and genre. The robustness of the model across these stimuli suggests a shared neural representation of the subjective experience of arousal across different situations, highlighting an intrinsic similarity in how the brain represents arousal from suspenseful moments in *Sherlock* to exciting moments in *Friday Night Lights*.

In addition, our research introduces a novel methodological approach for predicting moment-to-moment arousal ratings during movie watching. Collecting arousal ratings from human participants is both labor-intensive and time-consuming. Given the increasing availability of open fMRI datasets that include movie-watching data [74–76], our model offers a valuable tool for researchers who wish to obtain continuous measures of arousal without the need for additional human ratings. To that end, we have made the trained model weights and analysis scripts publicly available (see *Data and code availability*), such that other researchers can apply the weights to their own data or other open movie fMRI datasets.

In identifying a neural representation of emotional arousal that generalizes across datasets, we show that the brain represents arousal states similarly across different contexts. This finding potentially provides a neural basis for theories of emotion proposing that arousal states are inherently confusable and susceptible to contextual misattribution. In particular,

the two-factor theory of emotion proposes that emotional experiences arise from the combination of physiological arousal and the cognitive interpretation of that arousal [77]. To interpret arousal, individuals typically rely on contextual cues in the immediate environment. In cases where the source of arousal is ambiguous, people may mistakenly attribute their arousal to an incorrect source, a phenomenon known as the misattribution of arousal [78–81]. For example, a person who arrives at a job interview after climbing up a flight of stairs might mistakenly attribute the arousal from the physical activity to nervousness about the interview, which can in turn affect their performance on the interview. Our results suggest that such confusions may occur because the brain represents arousal states similarly across different contexts. This shared representation of arousal could serve as a fundamental substrate upon which emotional experience is constructed and interpreted [82–86].

Our connectivity-based predictive models complement prior work that predicts affective experience from multivariate activity patterns [27,69,72,73,87]. For example, Čeko and colleagues [73] demonstrated that negative affect could be predicted from distributed multivariate activity patterns, and that the predictive model generalized across negative affect induced by mechanical pain, thermal pain, aversive sounds and aversive images. More recently, work by Zhang and colleagues [87] predicted emotional arousal from whole-brain activity patterns, and showed that their predictive model also generalized across movies and individuals. Here, we extend these findings by demonstrating that emotional arousal is not only reflected in regional activity patterns but also in large-scale connectivity between brain networks. Specifically, our findings highlight that functional connections within and between multiple networks—including the ventral attention network, the frontoparietal network, the dorsal attention network, and the default mode network—were associated with continuous fluctuations in emotional arousal. This suggests that network interactions, in addition to regional activity patterns, plays a role in tracking emotional arousal over time, which implies that arousal may be encoded in the dynamic interactions between brain systems, aligning with network-based models of affect and cognition [45]. In this view, affective states emerge from the coordinated activity of multiple brain networks rather than being localized to discrete brain regions.

The positive associations between arousal and connections linking the DMN and FPN, as well as the DAN and VAN, indicate that arousal may be facilitated by interactions between systems involved in internally directed cognition [88], executive control [89], and attentional orienting [90]. These findings are consistent with theories proposing that arousal enhances neural flexibility, allowing for efficient coordination between higher-order cognitive systems to regulate emotional and attentional states [60,91–92]. Conversely, the negative association between arousal and DAN-VIS connectivity suggests that heightened arousal may relate to reduced coupling between attentional control and early sensory processing. This could reflect a shift in neural resources away from lower-level visual processing and toward internally guided or top-down attentional mechanisms. Future work will be needed to tease apart the functional contributions of these network interactions and determine how they differentially support arousal-related changes in cognition.

The use of publicly available datasets, where fMRI and affective ratings were collected from separate participant groups, reflects both a strength and a limitation of our study. A strength is that connectivity-based arousal models trained on one group of participants generalized effectively to another, suggesting that neural representations of arousal are consistent not only across different stimuli but also among individuals. This design allows for training and testing in new samples, enhancing the applicability of our model to the growing number of open movie fMRI datasets. However, as we did not train and test on participant-specific arousal data, the models would not pick up on idiosyncratic neural representations of arousal. While our results reveal generalized representations of emotional arousal, this does not rule out the coexistence of individual-specific representations [55,93]. Thus, incorporating individual-specific data in future studies could further enhance prediction accuracy.

Our study focused on predicting emotional arousal during dynamic, naturalistic stimuli, where affective experiences vary continuously over time. While this approach is valuable in that it mirrors how emotions develop and evolve in response to unfolding events, it is unknown whether the model would generalize to brief, momentary stimuli such as affective images [24,30] or sounds [3,4,25] that are also known to modulate emotional arousal. Testing this generalization

is beyond the scope of the current study, in part because our modeling approach relies on dynamic connectivity patterns calculated using a sliding window approach and is not well-suited for stimuli presented over very short durations (e.g., 2–4 seconds). However, we recognize this as an important avenue for future research. New methods for computing dynamic connectivity at finer temporal resolutions (e.g., [94]) could help extend the current modeling framework to capture emotional arousal in response to momentary stimuli.

An additional consideration is the complexity and heterogeneity of arousal as a psychological construct [35,95]. In a recent review, Satpute and colleagues [35] deconstructed arousal into three varieties: wakefulness arousal, typically measured using EEG desynchronization; autonomic arousal, typically assessed via physiological measures such as pupil dilation or skin conductance; and (3) affective (i.e., emotional) arousal, typically measured through self-report. Overlapping and distinct neural mechanisms are thought to underlie the three varieties of arousal. Our study focuses specifically on emotional arousal, and our use of self-report is consistent with prior studies. However, how emotional arousal relates to wakefulness and physiological arousal during movie watching is unclear. Indeed, prior work suggests different varieties of arousal may diverge [87,96,97], and it is unknown whether our predictive model would generalize to the other two types of arousal. Our approach of testing whether predictive models trained on one dataset generalize to others paves the way for future studies to explore whether models trained on one form of arousal can predict another. This line of research could reveal overlapping neural representations across different varieties of arousal.

In contrast to emotional arousal, we were not able to predict emotional valence from whole-brain dynamic functional connectivity. Null results are inherently difficult to interpret, but we offer several possible speculations. One reason could be that the experience of valence is idiosyncratic during movie-watching, in which case the group-average ratings from a different sample of individuals may not be representative of the experience of the individual participant. Thus, even if the model were successful at predicting the subjective experience of valence in a participant, the predicted output may not match the group-average rating time course of a different sample. Future work that collects subjective ratings and brain data from the same participants will be necessary to test this possibility.

Another possibility is that the neural representation of valence is more readily captured by multivariate activity patterns than by connectivity patterns. In line with this possibility, prior studies have successfully predicted the valence of stimuli, such as images, tastes, words, and movies, from activity patterns [27,28,72,73,93,29]. Alternatively, the neural representation of valence is context-specific, and thus does not generalize across diverse movies. Notably, prior studies that predicted valence often used brief, context-free stimuli (e.g., isolated images or words) that may constrain the subjective experience of valence. For instance, in the absence of context, viewing an image of a smiling baby may elicit a basic experience of pleasantness that would be similar to that when viewing an image of adorable puppies. In contrast, extended, narrative-driven stimuli, likely evoke richer and more nuanced emotional experiences [98]. Here, the positive feelings elicited by Sherlock's witty banter with Watson in one scene may carry a different qualitative and neural signature compared to positive feelings felt during a victory by the football team in *Friday Night Lights*. This interpretation aligns with a meta-analysis by Lindquist et al. [99], which found no consistent brain regions associated with monotonically increasing or decreasing valence across studies, further suggesting that valence representation may depend on the specific context.

In summary, our study identified a generalizable neural representation of arousal encoded in dynamic functional connectivity, but did not find a parallel generalizable representation of valence. The findings highlight the relationship between arousal states and the dynamics of large-scale functional networks. This work extends our understanding of how affective experience is encoded in the brain, and provides a methodological approach of probing the neural basis of affective experience in naturalistic contexts using functional neuroimaging and machine learning techniques. Our study paves the way for future studies investigating the relationship between everyday affective states from neural dynamics. By integrating our approach with deep learning methods that can extract stimulus features from multimodal sensory stimuli [100], future studies can explore the mechanisms by which specific aspects of a stimulus drive arousal responses and influence connectivity patterns.

## Methods

**Ethics Statement.** All experimental procedures were approved by the Institutional Review Board of the University of Chicago (Protocol # IRB22–0002). Participants provided informed written consent before the start of the study and were compensated for participation with cash or credits for class.

**Subjects.** One hundred and twenty individuals participated in the behavioral experiment (83 female, mean age 20.45<<Eqn1>> 2.15 years). Subjects were randomly assigned to four experimental conditions, namely, *Sherlock* valence, *Friday Night Lights* valence, *Sherlock* arousal, and *Friday Night Lights* arousal. Each condition includes data from 30 subjects. The rating time courses were significantly correlated across individuals in each condition (S1 Fig).

**Stimuli and experiment design.** A 48min 6s segment of the BBC television series *Sherlock* [47], and a 45min 18s segment of the NBC serial drama *Friday Night Lights* (FNL) [48] were used as the audio-visual movie stimuli in the study. The 30s audiovisual cartoon (*Let's All Go to the Lobby*) was removed from the original *Sherlock* movie stimuli. Both stimuli were divided into two segments (23min and 25min 6s for *Sherlock*, 23min 45s and 21min 33s for *FNL*). We prepended a 5s countdown video to the beginning of each segment to prepare participants to the beginning of the movie. Before rating the movie, participants performed a practice task where they rated valence or arousal of a short video from the movie *Merlin* [50] (2:03–3:10 for valence, and 5:13–6:19 for arousal) to familiarize themselves with the task.

Participants watched either *Sherlock* or *FNL*, and continuously rated either how positive or negative (valence conditions), or how aroused or not aroused (arousal conditions) the videos made them feel at each moment while watching the video. Participants who provided valence ratings were told that positive valence refers to when they were feeling pleasant, happy, excited, and negative valence refers to when they were feeling unpleasant, sad, angry. Participants who provided arousal ratings were told that high arousal refers to when they were feeling mentally or physically alert, activated, and/or energized, and low arousal refers to when they were feeling very mentally or physically slow, still, and/or de-energized. Participants pressed "left" or "right" keys on a keyboard to adjust a slider bar on a 1–25 scale. The two ends of the scale were labeled as "Very Negative" and "Very Positive" for participants in the valence conditions, and "Completely Not Aroused" to "Completely Aroused" for participants in the arousal conditions. Participants were encouraged to move the slider bar for even slight changes in how they felt. The button presses were recorded by a program coded in jsPsych (www.jspsych.org). Participants had the option to take a break after they watched the first video segment and continued to watch the second segment whenever they felt ready to do so. Participants completed a post-experiment survey after they finished the main rating experiment, where they reported the overall valence or arousal rating of the two segments, demographics and whether they had watched the video before. 15, 17, 3, and 5 participants had viewed the movie episode before the experiment in the *Sherlock*-valence, *Sherlock*-arousal, *FNL*-valence, *FNL*-arousal condition respectively. As approximately half of the participants had previously watched *Sherlock*, we checked if the reliability of the ratings differed between participants who had or had not previously watched the episode. There was no significant difference in the pairwise rating similarity of the two groups both for the arousal ($t(29) = 1.157$, $p = .257$) and valence ratings ($t(29) = 1.543$, $p = .134$). In addition, the arousal models trained on the behavioral ratings from participants who had (mean r = .234, p = .019; RMSE = .995, $p = .024$) and had not (mean r = .297, p = .008; RMSE = .964, $p = .004$) seen the movie generalized to *Friday Night Lights*.

**Movie analysis and annotations.** We extracted 7 low-level visual features (hue, saturation, pixel intensity, motion energy, presence of a face, whether the scene was indoor or outdoor, presence of written text) and 3 low-level auditory features (amplitude, pitch, presence of music) from both *Sherlock* and *FNL*. We computed hue, saturation, pixel intensity ('rgb2hsv'), audio amplitude ('audioread') and pitch ('pitch') in MATLAB [56] (R2022b, The Mathworks, Natick, MA), and motion energy (pliers.extractors.FarnebackOpticalFlowExtractor) in Python [101] (version 3.9). Additionally, author J.K. coded whether each video frame contained a face (presence of face = 1, absence of face = 0), written text (presence of written text = 1, absence of written text = 0), whether the scene was indoor or outdoor (indoor = 1, outdoor = 0), and whether music played in the background (presence of music = 1, absence of music = 0).

**Behavioral data analysis.** Considering that rating changes can happen at any time during the movie and the time points of the changes differ across subjects, we resampled the ratings to one rating per TR (1.5s for *Sherlock* and 2s for *FNL*). Resampled time courses of the two movie segments from the same subject were concatenated, and z-scored across time. To compute the intersubject correlation of the affect ratings, we computed the Pearson correlation between each individual's subjective rating timecourse to the group-averaged rating timecourse with this individual left out. We calculated the similarity within each condition (i.e., *Sherlock*-valence, *Sherlock*-arousal, *FNL*-valence, *FNL*-arousal). The pairwise *r*-values were then Fisher *z*-transformed and averaged across individuals. An inverse transform was then applied to the resulting average *z*-value to obtain the average mean intersubject correlation for a given condition. To test whether group-mean ISC significantly deviated above zero, we applied a nonparametric permutation approach (10000 permutations) where 50% chance of sign-flipping was applied to every subject's rating similarity before averaging into group-mean. The normalized rating time courses of all subjects within each condition were averaged, and the group-average time courses were treated as the ground-truth affective experience time-locked to the stimuli.

All subsequent fMRI analyses in the study involved the application of a tapered sliding window. In order to align the behavioral time courses to fMRI data time courses, we convolved the group-average time courses with the hemodynamic response function (HRF), and applied a tapered sliding window to the convolved behavioral time courses. We applied a sliding window size of 30 TR (45s) for *Sherlock* and 23 TR (46s) for *FNL*, and a step size of 1TR and a Gaussian kernel $\sigma = 3$TR. We followed Song et al. [36] in determining the weights for cropping the tail of the Gaussian kernel at the beginning and end of the time courses.

**MRI acquisition and preprocessing.** We acquired the raw structural and functional images of the *Sherlock* dataset from Chen et al. [47], and the *FNL* dataset from Chang et al. [48]. The *Sherlock* data were collected on a 3T full-body scanner (Siemens Skyra) with a 20-channel head coil. Functional images were acquired using a T2*-weighted echo-planar imaging (EPI) pulse sequence (TR/TE = 1500/28 ms, flip angle 64°, whole-brain coverage 27 slices of 4 mm thickness, in-plane resolution $3 \times 3$ mm$^2$, FOV $192 \times 192$ mm$^2$), with ascending interleaved acquisition. Anatomical images were acquired using a T1-weighted MPRAGE pulse sequence (0.89 mm$^3$ resolution). The *FNL* data were collected on a 3T Philips Achieva Intera scanner with a 32-channel head coil. Functional images were acquired in an interleaved fashion using gradient-echo echo-planar imaging with prescan normalization, fat suppression and an in-plane acceleration factor of two (i.e., GRAPPA 2), and no multiband (i.e., simultaneous multislice) acceleration (TR/TE = 2000/25 ms, flip angle 75°, resolution 3 mm$^3$ isotropic voxels, matrix size 80 by 80, and FOV $240 \times 240$ mm$^2$). Anatomical images were acquired using a T1-weighted MPRAGE sequence (0.90 mm$^3$ resolution).

We applied the same preprocessing pipeline to the *Sherlock* and *FNL* dataset, which included MCFLIRT motion correction, high-pass filtering of the data with a 100-ms cut-off, and spatial smoothing using a Gaussian kernel with a full-width at half-maximum (FWHM) at 5 mm. The functional images were resampled to 3 mm$^3$ isotropic space, and registered to participants' anatomical images (6 d.f.), then to the Montreal Neurological Institute (MNI) space using affine transformation (12 d.f.). Preprocessing was performed using FSL/FEAT v6.00 (FMRIB software library, FMRIB, Oxford, UK).

**Dynamic connectome-based predictive modeling.** We applied dynamic connectome-based predictive modeling, an approach introduced in Song et al. [36] and available at https://github.com/hyssong/NarrativeEngagement, to predict arousal and valence. We first extracted BOLD signals across the whole brain using the 114-ROI cortical parcellation scheme of Yeo et al. [53], along with the 8-ROI subcortical parcellation from the Brainnetome atlas (bilateral amygdala, hippocampus, basal ganglia, and thalamus), yielding a total of 122 ROIs. We averaged the blood-oxygen-level dependent (BOLD) time courses of all voxels in each ROI. For each dataset, the dynamic functional connectivity (FC) patterns were extracted from the ROI-based BOLD time courses using a tapered sliding window approach, where the Fisher's z-transformed Pearson's correlation between the BOLD time courses of every pair of ROIs were computed within each tapered sliding window (Window size - *Sherlock*: 30 TRs = 45s; *FNL*: 23 TRs = 46s). Hyperparameters of the sliding window for brain data were the same as those for behavioral data (Step size: 1TR, Gaussian kernel $\sigma = 3$TR). In the

complementary analysis where we examined how the predictive accuracy of the model would change over different timescales of dynamic functional connectivity patterns, we also used window sizes of 15s, 30s, 60s and 75s in addition to the original 45s or 46s in the main analysis. The same window sizes were also applied to the behavioral ratings to match the timescale of function connectivity patterns.

**Within-dataset FC prediction.** Within each dataset, we built separate CPMs to predict valence or arousal time courses using leave-one-subject-out cross-validation. Following prior CPM work [36,38,102], we perform feature selection using data from N-1 subjects. Specifically, in each round of cross-validation, we conducted feature selection, where functional connections (FC) that significantly correlated with affect time course (i.e., average behavioral valence or arousal ratings) in the training set of $N - 1$ participants were selected (one-sample Student's $t$ test, $p < .01$). A non-linear support vector regression model (python package svm.SVR from sklearn, https://scikit-learn.org/, kernel = "rbf") was trained in the set of $N - 1$ participants to predict the group-average behavioral time course from the selected FC features, and tested on the held-out participant. The predictive accuracy of each round of cross-validation was calculated as the Pearson's correlation between the predicted and ground-truth arousal time course. Correlation coefficients of all rounds of cross-validation were Fisher's z-transformed, averaged across all cross-validation folds, and transformed back to an average r-value as the final measure of model performance.

As the affect time course is the same between training and test subjects viewing the same movie, the correlation between predicted and observed affect dynamics is inflated. To overcome this potential issue and assess that predictive power is higher than what we would expect due to this inflation alone, we conducted nonparametric permutation tests where we compared model performance against null distributions generated by training and testing the model on phase-randomized behavioral ratings (1000 permutations). These null distributions provide a baseline measure of model performance when training and testing on the same behavioral measure, and allow us to assess the specificity of our results to the affect time course of interest. We assumed a one-tailed significance test, with $p = (1 +$ number of null $r$ values <<Eqn2>> empirical $r)/(1 +$ number of permutations).

**Across-dataset FC prediction.** The set of FCs selected in every round of cross-validation in within-dataset prediction was used as the features in across-dataset prediction. All subjects' data in the training dataset (e.g., *Sherlock*) were used to train a non-linear SVM to learn the mapping between the patterns of selected FC and the group-average affect time course. The trained model was applied to the held-out dataset, to predict each subject's valence or arousal time course from their FC patterns. The predictive accuracy and statistical significance were evaluated in the same way as within-dataset FC prediction. However, unlike the within-datasat FC prediction, training and test affect time course were different for across-dataset FC prediction because they were measured from different narrative stimuli. Above-chance prediction would thus indicate that the model generalizes across individuals and narratives, suggesting that the corresponding affect time course can be predicted from FC dynamics.

**Arousal networks.** For each dataset, we termed the set of functional connections that were selected in every round of cross-validation in predicting arousal as that dataset's arousal network. We defined FC features that positively, or negatively predict arousal as the high- or low-arousal networks, respectively. To examine the relationship between the arousal network and canonical macroscale functional networks in the brain, we grouped the 122 ROIs into 8 functional networks: the visual (VIS), somatosensory-motor (SOM), dorsal attention (DAN), ventral attention (VAN), limbic (LIM), frontoparietal control (FPN), default mode (DMN), and subcortical (SUB) networks. These networks were previously identified based on resting-state functional connectivity MRI of 1000 subjects [53]. The set of functional connections that occurred in both the *Sherlock* and *FNL* arousal networks were defined as the overlapping arousal network. We used two methods to calculate statistical significance of network overlap. One was using the hypergeometric cumulative density function, which returns the probability of drawing up to x of K possible items in n drawings without replacement from an M-item population [58]. This was implemented in MATLAB as: p = 1 – hygecdf(x, M, K, n), where $x$ refers to the number of overlapping edges, K and n refers to the number of edges in each of the two networks, and M refers to the total number

of edges (7381). The other was the Jaccard index [59], which measures the similarity between two sets by computing the ratio between the size of their intersection and the size of their union. It was implemented using sklearn.metrix.jaccard_ score (https://scikit-learn.org/stable/modules/generated/sklearn.metrics.jaccard_score.html). To test the significance of the jaccard score, we compared the actual Jaccard score with a null distribution, generated by shuffling the position of selected FCs in the brain network (keeping the same number of selected FCs) before calculating the Jaccard score. These two methods generated consistent results.

We calculated the proportion of selected FCs relative to the total number of possible connections between each pair of functional networks. To statistically assess whether particular functional networks are represented in the arousal network more frequently than chance, we computed the proportion of selected FCs among all possible FCs between each pair of functional networks. To assess the significance of these proportions, we compared the proportions with a null distribution of 10000 permutations. We generated the null distribution by the following steps. First, we shuffled the positions of selected FCs (e.g., 593 for *Sherlock* high-arousal network) in the whole FC networks (i.e., 7381 FCs). Second, we divided the FC networks into the 8 canonical macroscale functional networks. Third, we computed the proportions for every pair of functional networks following the same procedure. Finally, the significance of each proportion is computed by comparing the actual value with the null distribution of 10000 permutations using a one-tailed t-test (fdr-corrected).

**Engagement behavioral ratings and networks.** We acquired the engagement behavioral ratings of the *Sherlock* movie from Song et al., [36] GitHub repository (https://github.com/hyssong/NarrativeEngagement). We averaged individual rating time courses across subjects, cropped the 52 TRs of the two cartoon segments (1–26 TR, 947–972 TR) from the group-average engagement rating time course, and z-normalized the rating time course. We computed the Pearson correlation between the average engagement and arousal rating time courses. The significance of the correlation was assessed by comparing the actual correlation with a null distribution, generated by circular shifting the average engagement rating time course 1000 times.

We trained a CPM on the engagement time course, and used the CPM to predict the arousal time course following a leave-one-subject-out cross validation procedure. The engagement time course was smoothed with the same tapered sliding window size as the arousal time course and functional connectivity patterns. The predictive accuracy in each run of cross-validation was calculated as the Pearson's correlation between model predictions and the ground-truth arousal time course. We then performed a paired *t*-test to test if model accuracy in predicting arousal was different between the CPM trained on engagement and the CPM trained on arousal.

To examine whether the neural representations of engagement and arousal shared underlying functional networks, we compared our arousal network with the engagement network acquired from Song et al. [36]. This previous work used leave-one-subject-out cross-validation to predict attentional engagement from dynamic functional connectivity in the *Sherlock* dataset. The *Sherlock* engagement FC network was the set of FCs selected in every round of cross-validation in predicting engagement. We computed the arousal-engagement overlap network by taking the overlapping FCs between the *Sherlock* arousal FC network (439 positive and 154 negative FC features) and the *Sherlock* engagement *FC* network (583 positive and 102 negative FC features), resulting in a network with 287 positive and 74 negative FC features.

**Overlap arousal network predictions on two more fMRI datasets.** To examine whether the overlap network between the *Sherlock* and *Friday Night Lights* arousal networks encoded a generalizable arousal representation, we trained a model on the overlap arousal network and tested on two more fMRI datasets, *North by Northwest* and *Merlin*.

The movie stimuli of the *North by Northwest* dataset was a 14:49-min long segment of the movie *North by Northwest*, an American spy thriller directed by Alfred Hitchcock. The fMRI data were collected on a 3T Philips Ingenia scanner at the MRI Research Center at the University of Chicago as part of an ongoing two-session study collecting task, movie, and annotated rest data. Only the data where participants watched the *North by Northwest* clip were analyzed here. Structural images were acquired using a high-resolution T1-weighted MPRAGE sequence ($1.000\,mm^3$ resolution). Functional BOLD images were collected on a 3T Philips Ingenia scanner with a 32-channel head coil

(TR/TE = 1000/28 ms, flip angle 62°, whole-brain coverage 27 slices of 3 mm thickness, in-plane resolution 2.5 × 2.5 mm$^2$, matrix size 80 by 80, and FOV 202 × 202 mm$^2$). Functional scans were acquired during the movie segment in a single continuous run (951 TRs). The functional run included an extra 57 TRs after the video ends where participants were instructed to stare at the cross over a blank screen. These TRs were cropped from the end of each functional image to match the length of the movie stimuli. Data from 32 participants were included here. The movie stimuli of the *Merlin* dataset was a 25-min long episode from BBC's *Merlin* (a British fantasy adventure drama series). The fMRI data was acquired from openneuro. The fMRI preprocessing pipeline and dynamic functional connectivity analysis were the same as those of *Sherlock* and *Friday Night Lights*. Additional behavioral ratings on valence and arousal were collected on both *North by Northwest* and *Merlin* movie clips. The continuous rating paradigm was implemented in a single continuous run, separately for *North by Northwest* and *Merlin*. The same behavioral preprocessing and analysis pipeline as those of *Sherlock* and *Friday Night Lights* were performed for *North by Northwest* and *Merlin*.

The set of FCs in the overlap arousal network between the *Sherlock* arousal network and the *Friday Night Lights* arousal network was used as the input features in the current across-dataset prediction to test its generalizability. The data from both all subjects in *Sherlock* and all subjects in *Friday Night Lights* was used to train a non-linear SVM to learn the mapping between the patterns of selected FC and the group-average arousal time course. The trained model was applied separately to the *North by Northwest* and *Merlin* dataset, to predict each subject's arousal time course from their FC patterns. The predictive accuracy and statistical significance were evaluated in the same way as within-dataset FC prediction.

**Examining out-of-sample generalizability of valence predictions with additional training data.** To examine whether a CPM of valence could generalize with more training data, we trained a CPM on the combined data from *Sherlock* and *Friday Night Lights*. In the combined neural data, we selected the 2067 FCs that significantly correlated with valence as input features. We trained a non-linear support vector regression model to predict the valence time courses from the combined neural data of the selected FCs, and tested this trained model separately in the *North by Northwest* and *Merlin* datasets.

**Defining positivity and negativity.** To identify moments with positive valence (positivity), we took the raw behavioral valence ratings (range 1–25, 13 = neutral), averaged across participants rating the same movie, extracted the time segments where the group-averaged raw behavioral rating was above 13, and concatenated them into one single time course. We z-normalized and convolved the new rating time course with a HRF, and smoothed the new rating time course using a tapered sliding window (window size *Sherlock*: 30 TRs; *FNL*: 23 TRs). Moments with negative valence (negativity) were identified following the same procedure except that ratings that were below 13 were extracted from the group-average valence rating time course. The *Sherlock* movie had 1114 and 800 TRs of positivity and negativity respectively, and the *FNL* movie has 758 and 594 TRs of positivity and negativity respectively (S7 Fig).

To build predictive models to predict positivity and negativity from brain fMRI activity, we preprocessed the brain data to assess how they would be associated with the positivity and negativity rating time courses. We separately extracted the TRs corresponding to the behavioral positivity and negativity moments from the 122 parcels' activation magnitude time courses, and applied the same tapered sliding window approach as described above to extract the dynamic functional connectivity patterns.

## Supporting information

**S1 Fig. Leave-one-participant-out rating similarity as a function of the number of subjects.** The x-axis represents the number of subjects. In each condition, the number of subjects, *k*, increased from 2 to 30. Corresponding to the point *k* on the x-axis, *k* subjects were randomly selected from all subjects for 1000 times with replacement, where each time the Fisher's z-transformed group-average ISC was computed. The gray area represents the standard deviation of the

distribution of permutations. The group-average rating similarity stabilizes as the number of subjects increases, suggesting it is unlikely for the group-average behavioral rating to increase in precision with more subjects.
(TIFF)

**S2 Fig. Across-dataset model predictive accuracy on arousal as a function of sliding window size.** The x-axis represents the number of subjects. In each condition, the number of subjects, $k$, increased from 2 to 30. Corresponding to the point $k$ on the x-axis, $k$ subjects were randomly selected from all subjects for 1000 times with replacement, where each time the Fisher's z-transformed group-average ISC was computed. The gray area represents the standard deviation of the distribution of permutations. The group-average rating similarity stabilizes as the number of subjects increases, suggesting it is unlikely for the group-average behavioral rating to increase in precision with more subjects.
(TIFF)

**S3 Fig. Across-dataset model predictive accuracy on valence as a function of sliding window size.** Model predictive accuracy when training on *Sherlock* and testing on *Friday Night Lights* **(A)** and when training on *Friday Night Lights* and testing on *Sherlock* **(B)**. We extracted dynamic functional connectivity patterns from both *Sherlock* and *Friday Night Lights* using 5 different sizes of tapered sliding window: 15s, 30s, 45s (the original size in the main analysis), 60s and 75s, and tested each model on test data at each window size, resulting in 50 conditions (2 datasets x 5 window sizes at training x 5 window sizes at testing). None of the 50 conditions showed above chance predictive accuracy in predicting valence.
(TIFF)

**S4 Fig. Functional anatomy of the overlap network between arousal and engagement.** Engagement network acquired from Song et al., 2021 [36]. In each figure, the upper triangle represents the network which contains functional connections that positively correlate with both arousal and engagement. The lower triangle represents the network which contains functional connections that negatively correlate with both arousal and engagement. Each cell represents the proportion of selected FCs among all possible FCs between each pair of functional networks. Networks pairs with above-chance selected FCs are indicated with an asterisk (one-tailed $t$-test, fdr-corrected $p < .05$). **A. The overlap between the arousal network and engagement network in the *Sherlock* dataset.** The connections between DMN and FPN, between DMN and VIS, between DAN and FPN as well as connections within the DMN and FPN positively predicted both arousal and engagement. The connections between VIS and DAN, between VIS and FPN, between FPN and DMN, as well as connections within the FPN negatively predicted both arousal and engagement (FDR-corrected $P < .05$). **B. The overlap between the *Sherlock-Friday Night Lights* arousal network and *Sherlock-Paranoia* engagement network.** Only the connection between DMN and FPN positively predicted both arousal and engagement in this cross-dataset manner.
(TIFF)

**S5 Fig. Arousal CPM predicts group-average arousal rating in two additional datasets.** Model predicted time courses across participants watching the same movie was averaged. The arousal time course for both *North by Northwest* and *Merlin* was predicted from a model trained on the overlap arousal network from *Sherlock* and *Friday Night Lights*. This averaged predicted arousal time course (predicted arousal) significantly correlated with the group-average arousal rating from a separate group of individuals (observed arousal). Model-predicted arousal time courses also corresponded with the plot of each movie. The gray bands indicate the standard deviation of predicted arousal across participants at each time point. In *North by Northwest*, the most arousing moment predicted by the model occurred in the scene when the protagonist was being chased by a plane, while the least arousing moment occurred during a long conversation between characters; In *Merlin*, the most arousing moment predicted by the model occurred when the two main protagonists had a brawl

in a tavern, while the least arousing moment occurred when a nameless soldier walked towards a campfire. Images in the figures are blurred for copyright reasons; in the experiment, movies were shown at high resolution.
(TIFF)

**S6 Fig. Arousal CPM predicts group-average arousal rating in *Sherlock* and *Friday Night Lights*.** Model predicted time courses across participants watching the same movie was averaged. The arousal time course for *Sherlock* was predicted from a model trained from *Friday Night Lights*. The arousal time course for *Friday Night Lights* was predicted from a model trained from *Sherlock*. This averaged predicted arousal time course (predicted arousal) significantly correlated with the averaged group-average arousal rating from a separate group of individuals (observed arousal). The gray bands indicate the standard deviation of predicted arousal across participants at each time point. Model-predicted arousal time courses also corresponded with the plot of each movie. In *Sherlock*, the most arousing moment predicted by the model occurred in the scene when the Sherlock Holmes was figuring out clues near a dead body, while the least arousing moment occurred during a long conversation between John Watson and his therapist; In *Friday Night Lights*, the most arousing moment predicted by the model occurred when a star player was severely injured in the ball game, while the least arousing moment occurred when the coach and a player having a conversation. Images in the figures are blurred for copyright reasons; in the experiment, movies were shown at high resolution.
(TIFF)

**S7 Fig. Equivalence tests to assess the across-dataset predictions on valence.** Equivalence tests were performed to assess whether the across-dataset predictive accuracies in predicting valence fell within a predefined range around zero, which would suggest that they were not only statistically non-significant but also practically insignificant. We defined an equivalence interval of [-.100,.100], with the bounds determined based on a small effect size of $r = .100$. Each datapoint in the box plot represents the predictive accuracy of each round of cross-validation. The black horizontal lines show the 95% percent CI of the mean *r* value. The equivalence test was significant for both across dataset accuracies (*Sherlock - Friday Night Lights*: $p = .368$, *Friday Night Lights - Sherlock*: $p = .306$), indicating that the observed predictive accuracies were statistically indistinguishable from zero within the defined bound.
(TIFF)

**S8 Fig. Connectome-based predictive modeling on positive and negative valence. A** Participants' subjective affective experience of positivity (left) and negativity (right) fluctuates over time during naturalistic movie watching. **B.** Dynamic functional connectivity does not predict subjective feelings of positivity and negativity. CPM performance in predicting positivity and valence for within-dataset (the left panel) and between-dataset (the right panel). The y-axis represents the predictive accuracy, as measured by Pearson's correlation between the model predicted time course and the observed group-average time course. Each datapoint in the box plot represents the predictive accuracy of each round of cross-validation. The black horizontal lines show the Fisher-z transformed mean *r* value. The gray half-violin plots show the null distribution of 1000 permutations, generated by phase-randomizing the observed group-average before training and testing the models. \*: $p < 0.05$, \*\*: $p < 0.01$, n.s.: $p > 0.05$, as assessed by comparing the empirical mean predictive accuracy against the null-distribution.
(TIFF)

## Acknowledgments

We thank Janice Chen and colleagues for open sourcing the *Sherlock* dataset, Luke Chang and colleagues for open sourcing the *Friday Night Lights* dataset, and Asieh Zadbood and colleagues for open sourcing the *Merlin* dataset. We thank the MRI Research Center at the University of Chicago (RRID:SCR_024723) for assisting with the collection of the *North by Northwest* dataset. We thank Emily Finn for sharing code on calculating optical flow. We thank Yizhou (Louisa)

Lyu for collecting part of the behavioral rating data. We thank members of the Bainbridge, Leong, Rosenberg, Bakkour (BLRB) community, especially members of the Motivation and Cognition Neuroscience Lab and Cognition, Attention, and Brain Lab at the University of Chicago for their helpful feedback.

## Author contributions

**Conceptualization:** Jin Ke, Yuan Chang Leong.

**Data curation:** Jin Ke, Zihan Bai.

**Formal analysis:** Jin Ke.

**Funding acquisition:** Monica D. Rosenberg.

**Investigation:** Jin Ke, Hayoung Song, Zihan Bai, Monica D. Rosenberg, Yuan Chang Leong.

**Methodology:** Jin Ke, Hayoung Song, Monica D. Rosenberg, Yuan Chang Leong.

**Resources:** Jin Ke, Hayoung Song, Monica D. Rosenberg, Yuan Chang Leong.

**Software:** Jin Ke, Hayoung Song, Monica D. Rosenberg, Yuan Chang Leong.

**Supervision:** Monica D. Rosenberg, Yuan Chang Leong.

**Validation:** Jin Ke, Zihan Bai.

**Visualization:** Jin Ke.

**Writing – original draft:** Jin Ke, Yuan Chang Leong.

**Writing – review & editing:** Jin Ke, Hayoung Song, Monica D. Rosenberg, Yuan Chang Leong.

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
