## [Decision Letter · Decision Letter 0]

13 Jan 2025

PCOMPBIOL-D-24-01866

Predicting emotional arousal during naturalistic movie-watching from whole-brain dynamic connectivity across individuals and movie stimuli

PLOS Computational Biology

Dear Dr. Leong,

Thank you for submitting your manuscript to PLOS Computational Biology. After careful consideration, we feel that it has merit but does not fully meet PLOS Computational Biology's publication criteria as it currently stands. Therefore, we invite you to submit a revised version of the manuscript that addresses the points raised during the review process.

Please submit your revised manuscript within 30 days Mar 15 2025 11:59PM. If you will need more time than this to complete your revisions, please reply to this message or contact the journal office at ploscompbiol@plos.org. Please include the following items when submitting your revised manuscript:

We look forward to receiving your revised manuscript.

Kind regards,

Ming Bo Cai

Academic Editor

PLOS Computational Biology

Marieke van Vugt

Section Editor

PLOS Computational Biology

**Additional Editor Comments:**

Reviewers are overall happy with the current manuscript. However, some concerns remain. I believe most of them can be addressed relatively easily by the authors.

**Journal Requirements:**

Potential Copyright Issues:

- Please confirm (a) that you are the photographer of Figures 1, S5, and S6, or (b) provide written permission from the photographer to publish the photo(s) under our CC BY 4.0 license.

5) Please amend your detailed Financial Disclosure statement. This is published with the article. It must therefore be completed in full sentences and contain the exact wording you wish to be published. Please ensure that the funders and grant numbers match between the Financial Disclosure field and the Funding Information tab in your submission form. Note that the funders must be provided in the same order in both places as well.

**Reviewers' comments:**

Reviewer's Responses to Questions

**Comments to the Authors:**

Reviewer #1: Review context: I am currently joining the review process for this paper after one round of review/revision at PLOS Comp Bio was already completed. However, previously I reviewed this manuscript and a revision thereof at a different journal. I have read the latest round of review, and the authors responses to it.

Overall evaluation: In my initial review of this paper (elsewhere) I stated “Generally, I found this to be a very strong paper methodologically, and one that thoroughly investigates and provides a clear answer to its research question.” I think that this evaluation not only holds true, but that the paper has been considerably strengthened – both by addressing my and the other reviewers comments in that prior review process, and by the current review responses at PLOS Comp Bio. I believe it would make a valuable contribution to the literature in its current form.

Specific comments: Since my own comments on this paper were satisfactorily addressed in a prior review process, I will limit myself to commenting on one of the main issues which arose in which round of review: Two reviewers commented on perceived inconsistency between the MPM findings and other, prior decoding studies focused on valence and arousal. While I appreciate the difficulty of interpreting a null result, and the other reviewers’ desire for an internally consistent literature, I would also caution against requiring new results to be consistent with old ones. If the past decades of the psych reform effort have taught us anything, it is that even “established” results frequently do not replicate for a wide variety of reasons. Requiring new findings to replicate old ones can reify what are actually noisy, inconsistent effects into artificially ironclad patterns. I’m not saying that that is necessarily happening here with respect to valence/arousal decoding MPMs. However, given the many differences between the present study and the past studies which have observed accurate MPMs in this space, I don’t think that the inconsistency here is necessarily a negative indicator about the accuracy of the present findings. The authors have chosen to remove the MPMs, which I don’t think was necessary, but does render this point is somewhat moot anyway.

Reviewer #2: Ke and colleagues develop a predictive model mapping from dynamic functional connectivity to behavioral ratings of arousal during naturalistic movie-viewing. They show that this model generalizes across individual brains (leave-one-subject-out cross-validation) and, importantly, across quite different movies (different characters, events, themes, etc). I’m receiving a version of this manuscript from the journal that has already been reviewed and revised at least once; for example, it seems that the authors have removed a less-convincing non-connectivity-based predictive model based on the previous round of reviews. That seems fine to me; I read the current version of the paper prior to looking at the reviews/responses and it seemed coherent and complete on its own. Overall, the manuscript is well-written, the figures are excellent, the modeling and statistical analyses seem rigorous, and the authors include a number of useful control analyses to reinforce their conclusions. In sum, I think this manuscript is already in a very good place and essentially ready for publication. I provide a handful of relatively minor comments and suggestions below.

My main comment is that I think the authors should be careful about how they set up and frame their work in terms of “individual-specific” neural signatures or “generalization across individuals”. For example, in the Introduction (page 3), the authors write “One possibility is that these representations are specific to each situation and each individual” and ultimately contrast their findings with this possibility. But the authors are mapping individual brain activity onto a Y variable that’s a *consensus* measure of arousal collected in a *separate sample* of subjects. This setup is not designed to find individual-specific neural signatures of arousal, because you don’t have individual-specific behavioral ratings collected in the very same fMRI subjects “in the moment” while they watch the films. Put differently: in this setup, you either find a signature that generalizes across subjects or you don’t; you would need behavioral ratings in the scanning subjects if you wanted to provide positive evidence for individual-specific neural signatures of arousal. I think the authors’ current approach is perfectly fine and interesting enough in its own right; I would just encourage them to be careful in the writing not to inadvertently set up the reader to expect individual-specific results.

In setting up the predictive modeling results (page 8), you state “In every round of cross-validation we selected FCs that significantly correlated with arousal in the training set as features.” I suspect this line will tend to raise red flags for readers, whether justified or not. If I understand correctly, this feature selection strategy is fair (not double-dipping) because the edges are selected in the training set (N – 1 subjects); maybe just say this more explicitly? Or if there’s a precedent in the field for doing feature selection in this way, maybe cite that as well? I suspect the implication that each training fold will use a slightly different set of features may bother people (although these feature sets should be highly correlated due to the similarity between training sets in leave-one-out cross-validation). Anyway, even just a single additional sentence here could preempt readers from becoming suspicious.

I’m still not sure I fully “get” why the within-dataset null distributions (e.g. Figure 3) are so high. In general, I think the statistical approach is sound and this positively shifted null distribution will only make the tests more conservative. But any extra effort to build an intuition for why this is the case up front would be appreciated.

I’m curious whether the authors have tried using intersubject functional connectivity (ISFC; Simony et al., 2016) rather than within-subject functional connectivity to drive their predictive model. ISFC would presumably isolate the stimulus-driven component of connectivity (maybe a good thing?), but might effectively filter out arousal signals that are not very precisely synchronized to the stimulus (maybe a bad thing?), and I suspect would ultimately reduce the generalizability of the model across different stimuli. Anyway, I understand this would complicate things and I don’t think it’s necessary for the current manuscript—just wondering if the authors have any intuitions or anecdotal evidence on how ISFC would affect their model.

In Figure 2, it looks like the error bands around the mean arousal/valence ratings are fairly consistent across time. I’m wondering: have the authors looked at time points of the film where variance in ratings is low versus high? Again, I don’t think this is necessary for the current manuscript—just curious.

I appreciate the effort the authors have put into making their code and models publicly available—that’s great! (Looking forward to seeing the behavioral rating data released publicly as well.)

Page 4: you say “30 in each condition”—at first I was confused about what “conditions” you were referring to, but now I understand you mean 2 movies (Sherlock, FNL) x 2 ratings (arousal, valence); you might want to articulate conditions sooner or adjust the wording

Figure 3 caption: “for within-dataset (the left panel) and between-dataset (the right panel)” feels like it needs another noun at the end (e.g. “cross-validation”) or just take out the “for”

References:

Simony, E., Honey, C. J., Chen, J., Lositsky, O., Yeshurun, Y., Wiesel, A., & Hasson, U. (2016). Dynamic reconfiguration of the default mode network during narrative comprehension. Nature communications, 7(1), 12141.

Signed: Samuel A. Nastase

Reviewer #3: Ke and colleagues presented models for predicting subjective arousal ratings of a movie based on functional connectivity patters from fMRI data. Subjective arousal ratings and fMRI data were collected from two separate groups of participants. The models were tested on ‘unseen’ datasets. The work is interesting and, in my view, makes a valuable contribution to the field.

Key points:

The paper presents an impressive computational approach to affective representation, and I particularly appreciate the generalization across different datasets. However, the insights regarding the representation of arousal and its connection to theory remain unclear. I would like to see a more in-depth discussion of the theoretical implications of these findings, as well as a closer examination of the information conveyed by the connectivity patterns.

The two datasets used in the paper are unbalanced (statistical significance aside). For the Sherlock dataset, valence ratings are more consistent across raters compared to arousal ratings, but the fMRI data is based on a smaller number of participants. In contrast, the Friday Night dataset shows more consistent arousal ratings than valence ratings, and has a larger fMRI sample. Could the absence of valence results be due to this imbalance? This issue could be explored using the supplementary datasets, but unfortunately, the equivalent information is not available for the two additional datasets provided in the supplementary materials. Did the authors attempt to predict valence for these datasets as well? Are the results ‘symmetric’ across data sets (switching training and testing)?

Other points:

- The authors state: “Such stimuli may not capture the diversity and complexity of real-world emotional experiences, raising questions about whether the neural representations of valence and arousal identified under these controlled conditions would generalize across varied and dynamic real-life situations, and across different individuals encountering these contexts. … allowing us to capture moment-to-moment affective responses across a variety of situations.” However, the current study does not address this limitation either. Watching a movie is just one example of 'varied and dynamic real-life situations' and does not fully capture the complexity of such contexts.

- Figure 4C: The term 'actual arousal' is misleading; in the figure it refers to the averaged subjective measure derived from a group of individuals distinct from those who provided the fMRI data. Please use a more descriptive label instead. Same applies to Supplementary figures.

- Discussion: “In line with this possibility, prior studies have successfully predicted the valence of stimuli, such as images, tastes, and words, from activity patterns [29,30,73,74,89].” Valence has also been predicted for movies (Kim et al., 2020).

- Infographics for rating and fMRI data in Figure 1 are not intuitive. How N=120 participants are allocated to conditions is unclear from the figure.

The paper would benefit from careful editing.

- The abstract does not accurately reflect the content of the paper. It mentions that predictive models of both valence and arousal were built, but the text later creates confusion by focusing solely on the arousal model in the analysis. I believe this oversight occurred during the initial revision.

- It is not clear at this point in the introduction what ‘condition’ is in reference to: “and collected continuous valence and arousal ratings of the two episodes from a separate group of participants (N = 120, 30 in each condition).”

- Figure 1: How N=120 participants are allocated to conditions is unclear from the figure (is it 30 per rating per film?)

- ‘… correlations was significantly positive’ is awkward

- Figure 4C: what does the gray color represent?

**Have the authors made all data and (if applicable) computational code underlying the findings in their manuscript fully available?**

Reviewer #1: Yes

Reviewer #2: Yes

Reviewer #3: Yes

PLOS authors have the option to publish the peer review history of their article (what does this mean? ). If published, this will include your full peer review and any attached files.

**Do you want your identity to be public for this peer review?** For information about this choice, including consent withdrawal, please see our Privacy Policy .

Reviewer #1: No

Reviewer #2: **Yes: ** Samuel A. Nastase

Reviewer #3: No

**Figure resubmission:**
---

## [Decision Letter · Decision Letter 1]

25 Mar 2025

Dear Dr Leong,

We are pleased to inform you that your manuscript 'Dynamic brain connectivity predicts emotional arousal during naturalistic movie-watching' has been provisionally accepted for publication in PLOS Computational Biology.

Best regards,

Amy Kuceyeski

Academic Editor

PLOS Computational Biology

Marieke van Vugt

Section Editor

PLOS Computational Biology

Reviewer's Responses to Questions

**Comments to the Authors:**

Reviewer #2: The authors have adequately addressed my comments (and the comments of other reviewers as far as I can tell)—I'm satisfied, and happy to endorse this for publication.

**Have the authors made all data and (if applicable) computational code underlying the findings in their manuscript fully available?**

Reviewer #2: Yes

PLOS authors have the option to publish the peer review history of their article (what does this mean? ). If published, this will include your full peer review and any attached files.

**Do you want your identity to be public for this peer review?** For information about this choice, including consent withdrawal, please see our Privacy Policy .

Reviewer #2: **Yes: ** Samuel A. Nastase

---

## [Editor Report · Acceptance letter]

PCOMPBIOL-D-24-01866R1

Dynamic brain connectivity predicts emotional arousal during naturalistic movie-watching

Dear Dr Leong,

I am pleased to inform you that your manuscript has been formally accepted for publication in PLOS Computational Biology. Your manuscript is now with our production department and you will be notified of the publication date in due course.

With kind regards,

Anita Estes
